# miR-582-5p Is a Tumor Suppressor microRNA Targeting the Hippo-YAP/TAZ Signaling Pathway in Non-Small Cell Lung Cancer

**DOI:** 10.3390/cancers13040756

**Published:** 2021-02-11

**Authors:** Bowen Zhu, Mitheera V, Megan Finch-Edmondson, Yaelim Lee, Yue Wan, Marius Sudol, Ramanuj DasGupta

**Affiliations:** 1Laboratory of Precision Oncology and Cancer Evolution, Genome Institute of Singapore, A*STAR, Singapore 138672, Singapore; mitheera@stanford.edu; 2Department of Physiology, NUS Yong Loo Lin School of Medicine, National University of Singapore, Singapore 117593, Singapore; marius.sudol@mssm.edu; 3Laboratory of RNA Genomics and Structure, Genome Institute of Singapore, A*STAR, Singapore 138672, Singapore; wany@gis.a-star.edu.sg; 4Mechanobiology Institute, National University of Singapore, Singapore 117411, Singapore; mfinch-edmondson@cerebralpalsy.org.au (M.F.-E.); mbily@nus.edu.sg (Y.L.)

**Keywords:** miR-582-5p, Hippo, YAP (Yes-associated protein or YAP1), TAZ (Transcriptional co-activator with PDZ-binding motif aka WWTR1), NSCLC (Non-small Cell Lung Cancer)

## Abstract

**Simple Summary:**

Lung cancers lead cancer-related mortalities, with Non-Small Cell Lung Cancer (NSCLC) representing a substantial proportion of these cases. Perturbation of Hippo-YAP/TAZ signaling in NSCLC could be mainly attributed to post-transcriptional regulators since genetic alterations to the signaling pathway are known to be rare. In this study, we identified miR-582-5p as a novel, post-transcriptional regulator of Hippo-YAP/TAZ signaling. Our work revealed an inhibitory function of miR-582-5p on YAP/TAZ signaling in NSCLC cells, whereby the tumorigenic potential is diminished upon the overexpression of miR-582-5p. We also uncovered the regulation of miR-582-5p expression by YAP/TAZ, suggesting a potential feedback loop of YAP/TAZ signaling mediated by miR-582-5p. Mechanistically, we discovered NCKAP1 and PIP5K1C, regulators of actin polymerization, as novel and direct targets of miR-582-5p. Restoring miR-582-5p expression in NSCLC cells resulted in deficient F-actin, which potentially mediates the miR-582-5p-driven tumor suppression in a YAP/TAZ-dependent manner. Our findings underscore the anti-tumor function of miR-582-5p in NSCLC, positing its therapeutic potential in YAP/TAZ-driven lung cancers.

**Abstract:**

The Hippo-YAP/TAZ signaling pathway is an evolutionarily conserved signaling pathway involved in a broad spectrum of biological processes, including tumorigenesis. Whilst aberrant Hippo-YAP/TAZ signaling is frequently reported in various cancers, the genetic alterations of this pathway are relatively rare, suggesting regulation at the post-transcriptional level. MicroRNAs play key role in tumorigenesis by regulating gene expression post-transcriptionally. Amongst the cancer-relevant microRNAs, miR-582-5p suppresses cell growth and tumorigenesis by inhibiting the expression of oncogenes, including AKT3, MAP3K2 and NOTCH1. Given the oncogenic role of YAP/TAZ in solid tumors, we scrutinized the possible interplay between miR-582-5p and Hippo-YAP/TAZ signaling. Correlation analysis in NSCLC cells revealed a positive relationship between the expression of mature miR-582-5p and the proportion of phosphorylated YAP/TAZ. Intriguingly, YAP/TAZ knockdown reduced the expression of mature miR-582-5p but increased that of primary miR-582. Overexpression of miR-582-5p resulted in increased phosphorylation of YAP/TAZ with a concomitant reduction in cell proliferation and enhanced apoptosis. Mechanistically, we find that miR-582-5p targets actin regulators NCKAP1 and PIP5K1C, which may be responsible for the observed alteration in F-actin, known to modulate YAP/TAZ. We postulate that regulation of the actin cytoskeleton by miR-582-5p may attenuate YAP/TAZ activity. Altogether, this study reveals a novel mechanism of YAP/TAZ regulation by miR-582-5p in a cytoskeleton-dependent manner and suggests a negative feedback loop, highlighting the therapeutic potential of restoring miR-582-5p expression in treating NSCLC.

## 1. Introduction

Lung cancer is one of the most aggressive cancers [1]. In 2018, the World Health Organization (WHO) reported lung cancers as the most prevalent form of cancer and the most common cause of cancer-related death. To date, therapies aimed to control the progression of lung cancer have had limited successes due to poor early-diagnosis, resulting in metastatic progression and the development of drug resistance upon therapeutic interventions [2,3]. Histologically, lung cancers are classified into two types, namely small cell lung cancers and non-small cell lung cancers (NSCLC), the latter of which accounts for the large majority (~85%) of all lung cancer cases [4].

The Hippo-YAP/TAZ signaling pathway is evolutionarily conserved from *Drosophila* to mammals, playing key roles in numerous physiological and pathological processes, including organ size control, embryonic development and tumorigenesis [5,6]. For instance, aberrant Hippo-YAP/TAZ signaling has been associated with numerous solid tumors, including NSCLC [7,8]. Several upstream mechanical and biochemical cues, such as cell-cell contact, epithelial cell polarization and ligand-receptor interactions, regulate two downstream nuclear effectors, Yes-associated protein (YAP) and its paralog, WW domain-containing transcription regulator protein 1 (WWTR1, also known as TAZ) [6,8,9,10,11]. As transcriptional co-activators, the functions of YAP and TAZ are dependent on their subcellular localization. The activation of the Hippo pathway is tumor-suppressive, whereby the phosphorylation of key pathway kinases, MSTs and MAPKs, leads to phosphorylation of LATS1/2 kinases, which in turn phosphorylate YAP and TAZ at the residues S127/S397 and S89/S311, respectively. Phosphorylated YAP and TAZ are retained in the cytoplasm in complex with 14-3-3 and subsequently degraded via ubiquitin-mediated proteolysis [11,12]. In contrast, in the absence of active Hippo signaling, unphosphorylated YAP and TAZ proteins translocate into the nucleus, where they activate pro-proliferative and anti-apoptotic transcriptional programs with the induction of target genes, such as ANKRD1, CTGF, CYR61, and BIRC5, through the interaction with TEA domain-containing transcription factors (TEADs) [8,13,14]. Notably, actin cytoskeleton dynamics has been suggested as a key mediator of YAP and TAZ regulation, whereby filamentous actin (F-actin) has been shown to decrease the phosphorylation of YAP/TAZ, resulting in increased nuclear YAP/TAZ signaling [15,16,17]. In addition, F-actin-severing proteins, e.g., Cofilin and Gelosin, function as negative regulators of YAP/TAZ by increasing YAP/TAZ phosphorylation [16]. Altogether, YAP and TAZ confer oncogenic potential to tumor cells in the absence of tight regulation by the Hippo pathway [10,18] or via upstream modulation by actin cytoskeletal dynamics [9,15].

Hyperactivation of YAP and TAZ driven by genetic alterations and/or inactivation of the Hippo pathway is generally rare, suggesting that YAP/TAZ activity is predominantly regulated post-transcriptionally. MicroRNAs (miRNAs) are an established class of short non-coding RNAs known to regulate networks of genes and pathways governing development, physiology, and pathology. Genes encoding miRNAs are transcribed by RNA polymerase II to generate primary miRNA transcripts (pri-miRNAs), which then undergo processing by the DROSHA-DGCR8 complex to produce precursor miRNAs (pre-miRNAs) [19,20]. Upon exportation to the cytoplasm, pre-miRNAs are processed by the Dicer complex to yield mature miRNA duplexes. One of the strands from this duplex, known as mature miRNA, guides the RNA-induced silencing complex (RISC) to bind to mRNA transcripts and silence their expression [21,22]. For instance, miR-375 exerts a tumor-suppressive role by antagonizing several Hippo-YAP/TAZ components, such as YAP, TEAD4, and CTGF, in hepatocellular carcinomas, pancreatic, and gastric cancers [23,24,25]. In addition, the tumor-suppressor miRNA miR-9-3p has been reported to inhibit cell proliferation by targeting TAZ in hepatocellular carcinoma [26]. Therefore, cancer-relevant miRNAs are involved in regulating the expression of pivotal genes, such as the components of the Hippo-YAP/TAZ signaling pathway, to promote or suppress tumorigenesis.

Among the miRNAs that are functionally important during NSCLC tumorigenesis, a recent study revealed reduced levels of miR-582-5p in NSCLC patient samples and cell lines. This phenomenon was proposed to be attributed to the inhibitory role of miR-582-5p in targeting Notch1 [27]. Notch1 has been identified as a key activator of Notch signaling, a pathway that promotes malignancy and resistance to therapies in NSCLCs by directing cell fate decisions and conferring self-renewal capacity to tumor-initiating cells [28]. In addition, miR-582-5p has also been reported to inhibit tumorigenesis by targeting various oncogenic events through inhibiting AKT3 [29,30,31], Rab27a [32], and TGF-β-SMAD [33] in cancer models, supporting its function as a tumor suppressor. 

In this study, we observe a negative correlation between miR-582-5p expression and YAP/TAZ signaling activity in NSCLC cells. Overexpression of miR-582-5p in NSCLC cell lines inhibited YAP/TAZ activity by promoting the phosphorylation of YAP/TAZ. Functionally, restoration of miR-582-5p expression attenuated cell proliferation, colony formation, and abrogated F-actin, while increasing cell apoptosis. Mechanistically, we demonstrate that miR-582-5p directly inhibits NCKAP1 and PIP5K1C, two genes that can positively regulate YAP/TAZ activity. We therefore propose a mechanism whereby miR-582-5p increases YAP/TAZ phosphorylation and inhibits their transcriptional activity by targeting modulators of actin dynamics, hence suppressing YAP/TAZ-driven cell proliferation, tumorigenesis, and anti-apoptosis. Our data also show that the miR-582-5p expression is reciprocally regulated by YAP/TAZ, highlighting a potential negative feedback loop. Altogether, our findings indicate the potential of miR-582-5p as a therapeutic target in NSCLC.

## 2. Results

### 2.1. MiR-582-5p Expression Positively Correlates with the Phosphorylation Rate of YAP/TAZ in NSCLC Cell Lines

Given the important roles of both the Hippo-YAP/TAZ pathway and miR-582-5p in solid tumors, including NSCLCs, we questioned whether there is any interplay between these two factors. To address this, we first determined the expression of miR-582-5p in a panel of NSCLC cell lines using miRNA RT-qPCR. Variegated miR-582-5p expression was observed, with high miR-582-5p expression in A549 and H460, moderate in H647 and H358, and low in H1975, H661, H1299, and H226 cell lines (Figure 1A). Western blot was conducted to evaluate the corresponding protein levels of YAP and TAZ, as well as their phosphorylated forms in these cell lines (Figure 1B, Appendix A). Spearman Correlation analysis of the quantified protein and miR-582-5p expression was then performed (Figure 1C–F). There was no significant correlation between the expression of total YAP protein and miR-582-5p (Figure 1C). However, expression of miR-582-5p positively correlated with the relative phosphorylation rate of YAP, which indicated the inactivation of YAP (Figure 1D). Similarly, despite the moderate, albeit insignificant, correlation between miR-582-5p and TAZ expression, a more substantial and statistically robust correlation was observed between miR-582-5p expression and the relative phosphorylation rate of TAZ, which indicated the inactivation of TAZ (Figure 1E,F). Altogether, these results demonstrated an inverse correlation between the expression of miR-582-5p and YAP/TAZ activity in NSCLC cells, hinting at an underlying regulatory relationship.

### 2.2. miR-582-5p Overexpression Suppresses YAP/TAZ Transcriptional Activity

Given the positive correlation between miR-582-5p expression and the proportion of phosphorylated YAP/TAZ, we sought to investigate the regulatory interactions between miR-582-5p and the Hippo-YAP/TAZ signaling pathway. As shown in Appendix A and Appendix A, knockdown of YAP and TAZ efficiently reduced their protein expression in H1299 and H661 cells. Concordantly, RT-qPCR revealed significant reductions in YAP/TAZ transcriptional targets (ANKRD1, CTGF and CYR61) (Appendix A). Moreover, a notable decrease in CYR61 protein level was also observed (Appendix A, Appendix A), further confirming that siRNA-mediated knockdown of YAP and TAZ successfully blocked YAP/TAZ-driven gene transcription. The expression of mature miR-582-5p was also observed to be moderately reduced upon YAP/TAZ knockdown (Appendix A). Surprisingly however, the levels of primary-miR-582 (pri-miR-582) were markedly elevated (Appendix A). This indicated two possibilities: YAP/TAZ could either inhibit the transcription of pri-miR-582, or it could post-transcriptionally promote pri-miR-582 processing to its mature form.

The precise role of YAP/TAZ in modulating the miRNA-processing pathway remains unclear and somewhat controversial. Nuclear YAP has been previously reported by Mori and colleagues [34] to inhibit miRNA biogenesis via sequestering DDX17, a cofactor of the Drosha-DGCR8 complex, thus negatively regulating mature miRNA expression. Alternatively, YAP has been shown to promote miRNA processing and increase mature miRNA levels through the modulation of the LIN28-*let-7*-Dicer axis [35]. To eliminate the regulatory function of YAP/TAZ on the miRNA processing machinery, in order to influence miR-582-5p levels, H1299 and H661 cells were treated with siRNA against YAP/TAZ in addition to Drosha. As expected, silencing Drosha led to an accumulation of pri-miR-582, but unexpectedly, a concomitant knockdown of Drosha and YAP/TAZ further elevated pri-miR-582 expression (Appendix A). These results suggested YAP/TAZ’s involvement in miR-582-5p regulation to be upstream of Drosha, for example at the transcriptional level. Subsequently, we profiled a previously published dataset where miRNA expression was examined upon the overexpression of YAP or TAZ in a gastric cancer cell line [36]. We observed two categories of miRNAs responding in a polarizing manner (Appendix A). This finding revealed the effect of YAP or TAZ on miRNA expression to be miRNA-specific. Interestingly, miR-582-5p was amongst the miRNAs that were upregulated by both YAP and TAZ overexpression (Appendix A), supporting our finding that YAP/TAZ are drivers of miR-582-5p expression. To summarize, we observed that YAP and TAZ exert a positive role in mature miR-582-5p expression even though they appear to transcriptionally inhibit pri-miR-582 expression to some extent.

The observed positive regulatory role of YAP/TAZ in miR-582-5p expression contradicts their negative correlation in NSCLC cell lines, leading us to explore whether miR-582-5p functions in regulating YAP/TAZ signaling. In order to test this, we overexpressed miR-582-5p in H1299 and H661 cells (Appendix A), followed by western blot and RT-qPCR analysis to determine the activity of YAP/TAZ. In accordance with previously published results [27,29,30,32], overexpression of miR-582-5p reduced the expression of AKT and Notch proteins (Appendix A). We then proceeded to investigate whether miR-582-5p exerts its function in YAP/TAZ regulation by examining the total protein expression of YAP/TAZ as well as their phospho-protein levels, namely pYAP-S127 and pTAZ-S89. Consistent with our previous findings (Figure 1D,F), overexpression of miR-582-5p increased pYAP-S127 and pTAZ-S89 protein levels (Figure 2A–D, Appendix A). Subsequently, immunofluorescence staining was employed to examine the subcellular localization of YAP and TAZ. Upon overexpression of miR-582-5p, there was an significantly increased cytoplasmic localization of TAZ in H1299 and H661 cells (Figure 2E–H). Similarly, the nuclear-to-cytoplasmic shift in localization of YAP was apparent in H1299 cells (Figure 2E,F). However, YAP remained uninfluenced upon miR-582-5p overexpression in H661 cells (Figure 2G,H). It is known that phosphorylation of YAP and TAZ results in their sequestration in the cytoplasm by 14-3-3 proteins and the consequent down-regulation of their transcriptional targets [37]. RT-qPCR analysis revealed that overexpression of miR-582-5p in H1299 cells led to marked reductions of ANKRD1, CTGF, and CYR61 (Figure 2I). In H661 cells, ANKRD1 and CYR61 were consistently reduced, whereas CTGF expression remained unchanged (Figure 2J), which is likely due to the marginal alteration of YAP localization (Figure 2G,H). As expected, based on our previous results (Figure 2A–D), overexpression of miR-582-5p failed to significantly alter YAP/TAZ mRNA (Figure 2I,J).

To further investigate whether the miR-582-5p-mediated YAP/TAZ inhibition is dependent on YAP/TAZ phosphorylation, we performed a rescue experiment with a phosphorylation-resistant mutant of TAZ (TAZ S89A) in H1299 cells. Interestingly, overexpression of TAZ S89A in the presence of exogenous miR-582-5p restored the downregulation of YAP/TAZ target genes caused by miR-582-5p overexpression alone (Figure 2K,L, Appendix A). Taken together, these data reveal that miR-582-5p negatively regulates YAP/TAZ signaling by altering the phosphorylation status of YAP/TAZ thus increasing the cytoplasmic retention of YAP/TAZ, effectively attenuating YAP/TAZ transcriptional activity. 

### 2.3. Overexpression of miR-582-5p Inhibits Cell Growth, Induces Cell Apoptosis, and Compromises F-Actin Cytoskeleton

Recently, several reports have described the tumor-suppressive role of miR-582-5p in a variety of cancer models [29,30,31,32,33]. Furthermore, YAP and TAZ are potent oncogenes that drive tumorigenesis by promoting cell proliferation and inhibiting programmed cell death [6,12,14,18,37]. Therefore, we hypothesised that miR-582-5p might inhibit NSCLC tumorigenesis by modulating the pro-proliferative and anti-apoptotic functions exerted by YAP and TAZ. To investigate whether miR-582-5p can inhibit YAP/TAZ-driven phenotypes, miR-582-5p was overexpressed in NSCLC cell lines followed by a series of functional assays. As shown in Figure 3A,B, cell growth kinetics of H1299 and H661 cells were measured by a CellTiter-Glo cell-viability assay following the transfection of miRNA mimics. Overexpression of miR-582-5p in both cell lines resulted in significantly slower cell proliferation. Furthermore, overexpressing miR-582-5p in H1299 and H661 cells led to a marked increase of Caspase-3/7 activity (Figure 3C,D), indicating elevated cell apoptosis in both lines. This suggested that the slower cell growth caused by miR-582-5p could be attributed, at least partially, to an increased rate of cell apoptosis. In addition to cell proliferation, we conducted an in vitro colony formation assay to measure the tumorigenic potential of NSCLC cells upon overexpression of miR-582-5p. As shown in Figure 3E–H, overexpression of miR-582-5p significantly inhibited colony formation in both H1299 (Figure 3E,F) and H661 (Figure 3G,H) cells, further exemplifying the anti-tumor function of miR-582-5p. In summary, these results consistently corroborate the tumor-suppressive function of miR-582-5p in NSCLC cells.

In addition to the observed changes in cell proliferation, colony formation and cellular apoptosis, miR-582-5p overexpression also resulted in a “reduced-spreading” phenotype in H1299 and H661 cells (Appendix A). Confocal microscopy revealed compromised F-actin in H1299 and H661 cells following the overexpression of miR-582-5p (Figure 3I,J). Concomitantly, we also observed diminished pFAK-S397 levels, indicating reduced cell-substratum adhesion, attributable to miR-582-5p overexpression (Figure 3K and Appendix A). This suggested that miR-582-5p may be involved in the regulation of the actin cytoskeleton, thereby altering mechanical forces.

### 2.4. miR-582-5p Suppresses Hippo-YAP/TAZ Signaling by Targeting Actin Cytoskeleton Regulators, PIP5K1C and NCKAP1

YAP and TAZ are primarily regulated by canonical Hippo signaling as well as mechanical forces, the latter of which are known to be mainly influenced by the actin cytoskeleton status of a cell. Examination of the upstream Hippo pathway regulator LATS1 in H1299 and H661 cells revealed that the overexpression of miR-582-5p did not increase LATS1 phosphorylation (Appendix A, Appendix A), suggesting the accumulation of phosphorylated YAP/TAZ to be driven by a Hippo-LATS independent mechanism. Given the regulatory role of the actin cytoskeleton in YAP/TAZ modulation, we wondered whether the observed alterations in YAP/TAZ phosphorylation status mediated by miR-582-5p could be explained by altered cytoskeletal dynamics. To investigate this, we first obtained a list of predicted miR-582-5p targets from TargetScan [22] and overlapped them with a list of genes involved in regulating actin cytoskeleton (sourced from Wikipathways) [38]. This yielded nine common genes as candidate targets of miR-582-5p (Figure 4A). We then proceeded to validate the expression of these nine genes by RT-qPCR upon the overexpression of miR-582-5p. Expression of three genes (APC, PIP5KC1, and NCKAP1) were markedly diminished by miR-582-5p overexpression in both H1299 and H661 cell lines (Figure 4B,C). Since endogenous APC expression was relatively low, we proceeded to examine the protein expression of NCKAP1 and PIP5K1C. Western blot analysis revealed reduced protein levels of these two genes upon transfection of miR-582-5p (Figure 4D, Appendix A). Therefore, we identified NCKAP1 and PIP5K1C as candidate targets of miR-582-5p that might be responsible for the aforementioned phenotypes.

Subsequently, we sought to validate whether miR-582-5p targets PIP5KC1 and NCKAP1 through a direct miRNA-mRNA interaction. We constructed luciferase reporters with the sequences bearing the TargetScan-predicted miR-582-5p binding sites on the 3’UTRs of NCKAP1 and PIP5K1C downstream of the 3’ end of a firefly luciferase gene. Analysis of the luciferase reporter activity demonstrated two responsive NCKAP1 sites on position 9987-9994 and 16459-16465 (herein referred to as NCKAP1-1 and NCKAP1-8) and one site of PIP5K1C on position 2179-2186 (herein referred to as PIP5K1C-2) (Figure 4E). Disruption of miR-582-5p binding, via the generation of luciferase reporters with a three-nucleotide mutation in the seed sequences of NCKAP1-1, NCKAP1-8, and PIP5K1C-2, successfully rescued the relative luciferase activity, albeit to various extents (Figure 4F–H). We further verified the two sites which exhibited the best reduction and rescue, namely NCKAP1-1 and PIP5K1C-2, by calculating the minimum free energy (MFE) of miR-582-5p binding on these sites using RNAhybrid [39]. Notably, the MFEs of miR-582-5p binding to NCKAP1-1 and PIP5K1C-2 were comparable to that of the published miR-582-5p binding site on NOTCH1 (Appendix A) [27]. Therefore, we conclude that NCKAP1 and PIP5K1C are direct targets of miR-582-5p.

PIP5KC1 is indispensable during the formation of focal adhesions, an event that serves as a modulator of Integrin-FAK signaling, a pathway implicated in the activation of YAP [40,41,42]. NCKAP1 is a stabilizer of WASF3, the latter of which promotes actin polymerization. Loss-of-function of both genes have been implicated in blocking tumorigenesis and metastasis in prostate, breast and lung cancers [43,44,45]. Notably, correlation analysis from The Cancer Genome Atlas (TCGA) database illustrated positive correlations between NCKAP1 or PIP5K1C and YAP/TAZ targets, namely, CYR61 and CTGF, in the cohorts of NSCLC patients (Appendix A). This suggested that NCKAP1 and PIP5K1C may play positive roles in the regulation of YAP/TAZ and thus, potentially connect miR-582-5p and YAP/TAZ signaling.

To gain insight into the role of PIP5K1C and NCKAP1 in regulating YAP/TAZ-driven phenotypes, we performed siRNA-mediated knockdown of PIP5K1C and NCKAP1. Indeed, depletion of NCKAP1 or PIP5K1C led to notable reductions in cell proliferation rates (Figure 4I,J). Moreover, PIP5K1C knockdown led to a significant increase in relative Caspase-3/7 activity, which was not observed upon the knockdown of NCKAP1 (Figure 4K,L). This signifies that PIP5K1C, but not NCKAP1, may play an anti-apoptotic role in H1299 cells. The observed phenotypical changes are likely due to the perturbed YAP/TAZ activity, because the loss of NCKAP1 or PIP5K1C resulted in the concomitant downregulation of YAP/TAZ targets ANKRD1, CTGF and CYR61 (Figure 4M,N). Therefore, we conclude that NCKAP1 and PIP5K1C are novel and direct targets of miR-582-5p. Importantly, their potential influence in YAP/TAZ signaling suggests a mechanism where miR-582-5p modulates YAP/TAZ activity via a NCKAP1/PIP5K1C-mediated regulation of actin cytoskeleton (Figure 4O). 

## 3. Discussion

In this study, we report a negative feedback regulation of the Hippo-YAP/TAZ signaling pathway mediated by miR-582-5p. We initially observed a positive correlation between the expression of miR-582-5p and YAP/TAZ phosphorylation rate, the latter of which serves as a surrogate marker of cytosolic and inactive YAP/TAZ. This original finding led us to investigate the interplay between miR-582-5p and the Hippo-YAP/TAZ signaling pathway. Our loss-of-function assays in the NSCLC models and the published dataset using gastric cancer cells [36] suggested that YAP/TAZ is required for miR-582-5p expression. We further explored this notion by knocking down YAP/TAZ in a background of deficient pri-miRNA splicing, which was induced by silencing Drosha in NSCLC cell lines. This triple knockdown experiment resulted in a further accumulation of pri-miR-582 from silencing Drosha alone, suggesting that YAP/TAZ may also negatively regulate the pri-miR-582 at the level upstream of Drosha in the miRNA biogenesis pathway. Therefore, our data revealed the dual roles of YAP/TAZ in regulating miR-582-5p expression whereby YAP/TAZ holds the potential to repress pri-miR-582 expression but induce mature miR-582-5p expression (Figure 4O). This duality in YAP/TAZ regulation may have arisen to finetune the expression of miR-582-5p and to maintain homeostasis in the expression of the target genes regulated by miR-582-5p.

The role of YAP/TAZ in miRNA biogenesis has been previously reported; Mori and colleagues [34] have suggested the influence of YAP on miRNA expression, although in a different cell model. Their study discovered a negative role of YAP on miRNA biogenesis, whereby nuclear YAP inhibits the processing and eventual suppression of certain miRNA subgroups. However, our observations suggest that YAP/TAZ is required for the maturation of miR-582-5p, the mechanism of which remains unclear. Interestingly, Chaulk et al. have reported an alternative mechanism in which YAP and TAZ indirectly promote Dicer expression and hence, miRNA maturation in a *Let-7*- and LIN28-dependent manner [35]. These discrepancies could be attributed to the choice of cell line models that may possess differences in miRNA biogenesis. Moreover, differential YAP/TAZ expression and localization may confer additional layers of complexity to such model systems. Nonetheless, future studies could focus on delineating the precise mechanism by which YAP and TAZ are involved in miR-582-5p regulation.

Remarkably, restoring the expression of miR-582-5p by transfection of miRNA mimics in cell lines that express relatively low levels of this miRNA enhanced the phosphorylation of YAP and TAZ, consequently inhibiting YAP/TAZ-mediated gene transcription. Subsequently, we investigated the biological functions of miR-582-5p by performing phenotypical assays to examine certain features in NSCLC cell lines that are associated with oncogenic transformation, including pro-proliferation, anti-apoptosis and ‘stemness’. As expected, overexpression of miR-582-5p consistently inhibited these phenotypes in H1299 and H661 cells. These observations were in accordance with the previously reported tumor suppressive roles of miR-582-5p in a number of cancer models [27,29,30,31,32,33]. In addition, we provided evidence to show the association between miR-582-5p and the Hippo-YAP/TAZ signaling pathway using a non-phosphorylatable TAZ mutant, namely TAZ S89A, whereby the overexpression of this mutant rescued the expression of YAP/TAZ target genes in the cells overexpressing miR-582-5p. This suggests that miR-582-5p-mediated regulation of YAP/TAZ is indeed dependent on their canonical phosphorylated sites. Given the critical role of the YAP/TAZ-driven transcriptome in tumorigenesis, it is likely that TAZ S89A and YAP S127A overexpression may exert a dominant function and make the cells resistant to miR-582-5p overexpression. However, systematic functional studies, including in vitro and in vivo assays, could be employed to further validate the observations in our study. 

Previous studies have established the actin cytoskeleton’s involvement in the regulation of YAP/TAZ activity in a Hippo-LATS independent manner [9,16]. In agreement with these findings, we did not observe a concomitant upregulation of LATS1 activity upon miR-582-5p overexpression when examining its phosphorylation status. Despite the lack of LATS activation, exogenous miR-582-5p still promoted the phosphorylation of YAP/TAZ and their consequent cytoplasmic retention. Similarly, the rescue YAP/TAZ transcriptional targets with the TAZ S89A mutant in the presence of exogenous miR-582-5p confirmed that miR-582-5p mediates YAP/TAZ inhibition in a phosphorylation-dependent manner. A possible explanation for this observation could be the F-actin-driven YAP/TAZ regulation that has been shown to be independent of the Hippo-LATS axis [9,15,16]. Moreover, we noted changes in cell morphology and altered F-actin in response to miR-582-5p overexpression. These findings led us to posit the potential role of miR-582-5p in influencing actin cytoskeleton dynamics, thereby regulating YAP/TAZ phosphorylation, their subcellular localization and transcriptional activity. In our search for potential miR-582-5p targets that also function as actin cytoskeleton regulators, we identified NCKAP1 and PIP5K1C as novel and direct targets of miR-582-5p. We showed that miR-582-5p could directly antagonize the 3’ UTR of NCKAP1 and PIP5K1C to repress their expression. Furthermore, diminished expression of NCKAP1 and PIP5K1C led to a reduced transcriptional activity of YAP/TAZ. These findings potentially serve as regulatory nodes between miR-582-5p and YAP/TAZ signaling. NCKAP1 has been recognized as a regulator of the WASF3 complex, which promotes actin polymerization [43,45]. Similarly, PIP5K1C promotes cell-substrate adhesion [42] in order to facilitate focal adhesion and F-actin formation. F-actin formation has been implicated in YAP/TAZ nuclear localization and transcriptional activation, hence exerting a positive role in YAP/TAZ-driven gene transcription [15,16,17]. Altogether, in light of previous published findings and the results from this study, we propose a putative function for miR-582-5p in the regulation of NCKAP1 and PIP5K1C, which in turn influences the actin cytoskeleton, and consequently the activity/localization of YAP/TAZ (see schematic of working model in Fig. 4O). Given the fact that miRNAs usually function through targeting multiple genes, overexpression of PIP5K1C and NCKAP1 either individually or concomitantly could demonstrate whether these two targets are sufficient to rescue the phenotypes driven by miR-582-5p, such as actin cytoskeleton dynamics, phosphorylation rates of YAP/TAZ and reduced YAP/TAZ transcriptional activity.

Altogether, the conclusions from our study are in consensus with previous studies exemplifying the tumor-suppressive nature of miR-582-5p via targeting known oncogenes in diverse cancer models. Moreover, our results suggest an actin cytoskeleton- and phosphorylation-dependent regulation of YAP/TAZ activity by miR-582-5p. These findings open new avenues for the exploration of miRNA-mediated therapeutic strategies to treat YAP/TAZ-driven NSCLC.

## 4. Materials and Methods

### 4.1. Cell Culture

Human NSCLC lines H460, H661, H647, H358, H1975, H661, H1299 and H226 were cultured in RPMI-1640 media (Gibco, Thermo Fisher Scientific, Waltham, MA, USA, Cat#11875101) with the supplementation of 100 U/mL Penicillin-Streptomycin (Gibco) and 10% fetal bovine serum (Gibco, Cat#15070063). A549 cells were cultured in high glucose DMEM media (Thermo Fisher Scientific, Cat#11965092) and supplemented with 1 mM Sodium pyruvate (Gibco, Cat#11360070), 100 U/mL Penicillin-Streptomycin and 10% fetal bovine serum. All cells were originally obtained from American Type Culture Collection (ATCC, Manassas, VA, USA) and cultured under 5% CO_2_ at 37 °C.

### 4.2. Cell Transfection

7.5 × 10^4^ H1299 cells and 1.0 × 10^5^ of H661 cells were plated in 12 well plates for 24 h before transfection of 10 nM hsa-miR-582-5p mimic (miR-582-5p) and mimic negative control (mimic Neg) oligonucleotides (Thermo Fisher Scientific, Cat#4427975) or siRNA against YAP/TAZ/Drosha/PIP5K1C/NCKAP1 and siRNA control (siRNA purchased from Ambion, Invitrogen, Thermo Fisher Scientific) using a lipo3000 transfection reagent (Invitrogen, Cat#L3000001) in accordance with the manufacturer’s instruction.

### 4.3. RNA Isolation and Quantitative Real-Time PCR (RT-qPCR)

RNA was isolated using RNeasy Mini Kits (QIAGEN, Hilden, Germany, Cat#217004) from adherent cell cultures according to the manufacturer’s instruction where the total RNA content was measured using NanoDrop. A total of 2 µg RNA was reverse transcribed using SuperScript IV VILO cDNA Synthesis Kit (Invitrogen, Cat#11756500). The resultant cDNA was diluted 10 times using Nuclease-free water before being used in RT-qPCR. RT-qPCR was executed on the ABI-7300 instrument (Applied Biosystems, Thermo Fisher Scientific) using KAPA SYBR FAST qPCR kits (KAPA Biosystems, Roche, Basel, Switzerland, Cat#KK4601) and gene-specific primers (Table 1) in a reaction volume of 10 mL. Relative gene expression was quantified using the 2^−ΔΔCT^ method with 18S ribosomal RNA (rRNA) serving as an internal control.

### 4.4. Quantification of miR-582-5p Expression

Mature miR-582-5p levels were measured using a Reverse Transcription-real time PCR (RT-qPCR) TaqMan® miRNA Assay (Thermo Fisher Scientific, Cat#4427975). 18SrRNA levels were used to normalize relative mR-582-5p expression whereby the primer and probe sequences were designed in-house and bought separately (Table 2). MiR-582-5p quantification was performed mostly in accordance with the manufacturer’s instruction where isolated RNA was reverse transcribed with a Taqman MicroRNA Reverse Transcription Kit (Applied Biosystems, Cat#4366596) and the qPCR step was performed with the Taqman miRNA assay described above. The only deviations from the manufacturer’s instructions were the use of 20 ng RNA for reverse transcription instead of the suggested 10 ng and usage of 4 mL cDNA for qPCR instead of the recommended 2 mL. The formula: 2^-deltaCt^ was used to calculate the relative expression of miR-582-5p to 18S. The delta Ct in the formula was computed using the following algorithm: delta Ct = Ct_miR-582-5p_ − Ct_18S_

### 4.5. Western Blot Analysis

Cultured cells were washed with phosphate-buffered saline (PBS) and subsequently lysed with RIPA buffer (Thermo Scientific, Thermo Fisher Scientific, Cat#89901) supplemented with 1x Protease inhibitor (Roche, Cat#11697498001) and 1x Phospho-STOP (Roche, Cat#04906845001). The resultant protein solution (10–20 mg) was separated by electrophoresis with the use of a 10% SDS –polyacrylamide gel. Utilizing wet transfer, size-separated proteins were then transferred onto a poly-vinylidene difluoride (PVDF) membrane (Bio-Rad, Hercules, CA, USA), which was subsequently blocked with 5% BSA dissolved in TBST (Tris Buffered Saline with the addition of 0.1% Tween 20) for 1 h at room temperature. Membranes post-blocking were incubated in diluted primary antibody solutions (Table 3) at 4 °C overnight. After primary antibody staining, membranes were incubated with either a conjugated IRDye 680RD (1:5000) or IRDye 800CW (1:5000) secondary antibody over a duration of 1 h at room temperature. After secondary antibody incubation, fluorescence was detected from these membranes using a LI-COR ODYSSEY scanner (LI-COR Biosciences, Lincoln, NE, USA).

### 4.6. Immunofluorescence (IF) Staining and Confocal Microscopy

A total of 4 × 10^4^ H661 cells and 2.5 × 10^4^ H1299 cells were seeded on glass slides in 24-well plates slides and allowed to grow for two days before fixation. After a cold PBS wash, cells on the slides were fixed with cold 3.7% formaldehyde (diluted in PBS) for 15 min and permeabilized in PBTX (PBS with the addition of 0.2% Triton-X100) for 10 min. Fixed cells were blocked with 1% BSA, which was diluted in PBST (PBS supplemented with 0.1% Tween 20), for 1 h at room temperature and incubated with YAP (diluted at 1:200; Cell Signaling Technology, Danvers, MA, USA, Cat#14074) and TAZ (diluted at 1:200; BD Biosciences, Franklin Lakes, NJ, USA, Cat#560235) primary antibodies overnight in 4 °C conditions. Subsequently, cells were stained with secondary antibodies (1:1000; Alexa 488/594—Life Technologies, Thermo Fisher Scientific) in combination with DAPI (1 drop of NucBlue in 500 mL; Invitrogen, Cat#R37606) and Phalloidin (1:50, Alexa 594) for a duration of 2 h at room temperature to detect nuclei and F-actin, respectively. Post-secondary antibody staining, cells were washed three times with PBST before being imaged under a Leica confocal microscope (Buffalo Grove, IL, USA). Images were analyzed using Image J to quantify the total intensity of nuclear and cytoplasmic YAP or TAZ per cell.

### 4.7. Cell Proliferation Analysis

A number of 2500–3000 treated cells were used for this assay and reseeded into 96-well plates on day 0 with supplementation of fresh media. The proliferation kinetics of these cells were measured with a CellTiter Glo reagent (Promega, Madison, WI, USA, Cat#G7572), following the manufacturer’s instructions. Luminescence emitted was measured with a Tecan microplate reader (The Infinite M1000). The relative proliferative index was computed by normalizing the readings of subsequent days to that of day 1.

### 4.8. Cell Apoptosis Analysis

Cells for this experiment were plated as described in Section 4.7. After 48 h from seeding, luminescence readings from the Caspase-3/7 assay and cell viability were measured as described in Section 4.7, upon the addition of the Caspase-Glo 3/7 assay reagent (Promega, Cat#G8092) and CellTiter Glo reagent, respectively. The relative rate of cell apoptosis was approximated using a ratio of Caspase-Glo 3/7 values against that of CellTiter Glo.

### 4.9. Colony Formation Analysis

Transfected cells from Section 4.2 were re-plated as single cell suspensions of 500–1000 cells per well in a 6-well plate. Fresh media was added every four days post-seeding for a duration of 10–14 days. After the incubation, Crystal Violet staining was performed as previously described [46]. Using ImageJ, the number of stained colonies per well was quantified in accordance to the following steps: Open file—Select area—Process—Binary—Watershed- analyze particles (size: 100–10,000 pixel^2^, circularity: 0.25–1.0).

### 4.10. Database Analysis

TargetScan was utilized to gather the predicted targets of hsa-miR-582-5p and to delineate the potential binding sites of hsa-miR-582-5p on the 3’UTR of NCKAP1 and PIP5K1C [22]. Wikipathways was used to compile the list of genes involved in the regulation of actin cytoskeleton [38]. RNAhybrid was employed to predict the minimum free energy [39] of miR-582-5p binding on sites present in 3’UTRs of NCKAP1, PIP5K1C, and NOTCH1. Correlation data were imported into R environment in order to plot the respective correlations between the relative expression of miR-582-5p and YAP, TAZ, pYAP, or pTAZ [47]. The TCGA correlation analyses of NCKAP1 or PIP5K1C and YAP/TAZ target genes were obtained from the GEPIA web server [48].

### 4.11. Plasmid Construction

For constructs used in the UTR analysis, the fragments (30~35 bp) containing the predicted miR-582-5p sites on the 3’UTRs of NCKAP1 or PIP5K1C and their mutants were obtained by annealing a pair of synthetic oligos (Table 4), resulting in the insertion of miR-582-5p target region in a 5′-3′ orientation downstream of the firefly luciferase gene in the pmiRGLO plasmid (Promega, Cat#E1330). For TAZ S89A overexpression, TAZ was amplified from the pBABE-hygro-hTAZ construct made in-house with the restriction sites, BamH I and Xho I, and subcloned into the pLenti-CMV-GFP-puro plasmid (a gift from Eric Campeau & Paul Kaufman, Addgene plasmid #17448) digested by restriction enzymes, Bam HI and Sal I. The recombinant plasmid was then subjected to mutagenesis (Q5 Site-Directed Mutagenesis Kit, New England Biolabs, Ipswich, MA, USA, Cat #E0554S) according to the manufacturer’s instruction in order to change the 89th amino acid in TAZ protein from Serine to Alanine. Oligo information can be found in Table 4. All plasmids have been verified by Sanger sequencing

### 4.12. Lentiviral Packaging and Viral Transduction

The lentiviral plasmids (1.6 µg) were co-transfected with three packaging plasmids (gifts from Didier Trono): pMDLg/pRRE (0.8 µg), pMD2.G (0.4 µg), and pRSV-Rev (0.4 µg), into HEKH239T cells (1.0 × 10^6^) plated in 6 cm dishes using Lipofectamine 3000 transfection reagent (Invitrogen, Cat#L3000150) according to the manufacturer’s instructions. The media was replaced with 3 mL of fresh media 24 h post-transfection. The viral supernatant was collected and filtered through a 0.45-micron filter 72 h after transfection. A total of 200–400 µL of the filtered supernatant was supplemented with 5 mg Polybrene (Santa Cruz Biotechnology, Dallas, TX, USA, Cat#sc-134220) and used to infect cells in 12-well plates while seeding.

### 4.13. Luciferase Assays

A total of 5 × 10^3^ H1299 cells and 8 × 10^3^ H661 cells were seeded in wells of a 96-well plate and incubated for 24 h before the concomitant transfection of 20 ng pmiR-GLO luciferase plasmids and 1 mL of 1 µM miRNA mimics as described in Section 4.2. One day post-transfection, wells were replenished with 50 mL of fresh RPMI media. Luminescence was read, as described in 4.7, at 48 h post-transfection after the addition of the Dual-Glo assay (Promega, Cat#E2920) reagents in accordance with the manufacturer’s protocol.

### 4.14. Statistical Analysis

All data presented are an average of 3–6 independent experiments, unless specified otherwise. The error bars denote standard deviation. The *p*-values were calculated using a two-tailed Student’s t-test to indicate the statistical significance of results yielded when transfection was compared to the controls and *p* < 0.05 *, *p* < 0.01 **, *p* < 0.001 *** respectively.

## 5. Conclusions

We delineated a negative regulatory function of miR-582-5p on YAP/TAZ-signaling in NSCLC cell lines in vitro. Examination of miR-582-5p levels indicated its positive correlation with YAP/TAZ phosphorylation rate in NSCLC cell lines. Overexpression of miR-582-5p in NSCLC cells revealed a significant reduction in YAP/TAZ-driven cell proliferation, apoptotic resistance and clonogenicity. Together with the observed morphological and actin cytoskeleton alterations, we deduced that the perturbed tumorigenic features upon miR-582-5p overexpression could be attributed to compromised YAP/TAZ activity in an actin cytoskeleton-dependent manner. In silico analysis identified NCKAP1 and PIP5K1C as novel miR-582-5p target genes and we demonstrated the direct binding of miR-582-5p to NCKAP1 and PIP5K1C transcripts. Concurrently, our results also revealed a YAP/TAZ-mediated inhibition of pri-miR-582 expression and induction of mature miR-582-5p expression, suggesting a potential feedback loop between miR-582-5p and YAP/TAZ transcriptional activity. Therefore, our study discovered a novel inhibitory role exerted by miR-582-5p in modulating YAP/TAZ activity by targeting NCKAP1 and PIP5K1C due to the potential disruption in actin-mediated YAP/TAZ activation. This work serves to underline the therapeutic potential of a potent tumor suppressor, miR-582-5p, in NSCLC.

## Figures and Tables

**Figure 1 cancers-13-00756-f001:**
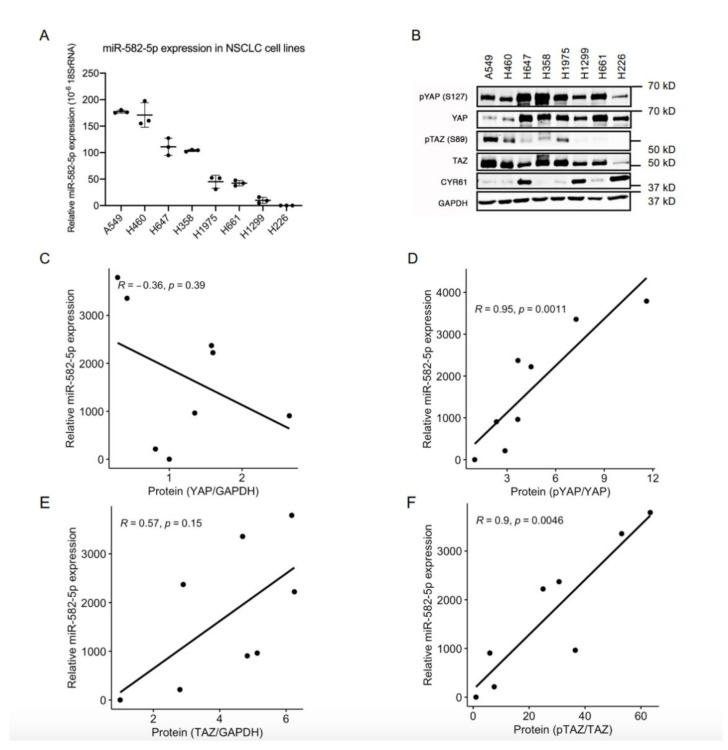
MiR-582-5p expression and YAP/TAZ phosphorylation rate are negatively correlated in NSCLC cells. (**A**) MicroRNA RT-qPCR analysis was performed on total RNA isolated from eight NSCLC cell lines to measure miR-582-5p levels. The ΔCT of miR-582-5p was normalized against that of 18SrRNA to determine its relative expression. (**B**) Western blot was conducted in eight NSCLC cell lines to measure the expression levels of YAP, TAZ, and their phosphorylated counterparts, as well as a YAP/TAZ transcriptional target, CYR61. GAPDH served as a loading control. (**C**–**F**) Dot plots represent the correlations between miR-582-5p expression and the expression of YAP (**C**), TAZ (**E**), or their phosphorylation rates, namely pYAP/YAP (**D**) and pTAZ/TAZ (**F**). The correlation coefficient (*R*-value) and the significance (*p*-value) were obtained by performing Spearman correlation analysis. The trendlines for the correlations were computed using R. Results are presented as the mean of triplicate experiments with error bars representing standard deviation.

**Figure 2 cancers-13-00756-f002:**
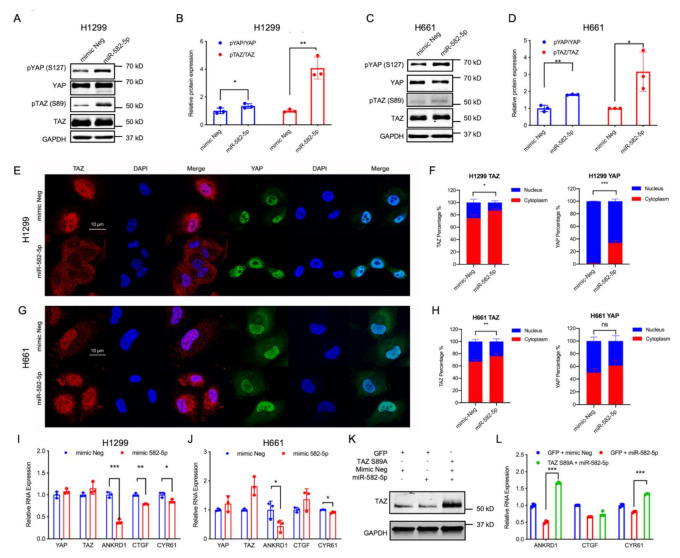
Overexpression of miR-582-5p attenuates YAP/TAZ signaling activity. (**A**–**D**) Western blot (**A**,**C**) and Image J quantification (**B**,**D**) were employed to measure the changes in protein levels of YAP, TAZ, and their phosphorylated counterparts following the overexpression of miR-582-5p in H1299 (**A**,**B**) and H661 (**C**,**D**) cell lines. GAPDH served as a loading control. (**E**–**H**) Representative images of immunofluorescence staining of YAP, TAZ, and DAPI, and the corresponding quantification for the subcellular localization of TAZ and YAP post-transfection of miR-582-5p in H1299 (**E**,**F**) and H661 cells (**G**,**H**). Scale bar = 10 µm. (**I**,**J**) RT-qPCR analysis was conducted upon overexpression of miR-582-5p in H1299 (**I**) and H661 (**J**) cells to examine relative mRNA levels of YAP, TAZ and the YAP/TAZ transcriptional targets, ANKRD1, CTGF and CYR61. The ΔCT of examined genes were normalized against that of 18SrRNA in order to determine relative expression. (**K**) Western blot analysis of TAZ protein levels upon concomitant overexpression of GFP/TAZ S89A and miR-582-5p mimics in H1299 cells. (**L**) RT-qPCR analysis of YAP, TAZ, and the YAP/TAZ transcriptional targets, ANKRD1, CTGF, and CYR61, upon concomitant overexpression of GFP/TAZ S89A and miR-582-5p mimics in H1299 cells. Results are expressed as the mean of triplicate experiments with error bars representing standard deviation. Statistical analysis was conducted using Student’s t-test with * *p* < 0.05, ** *p* < 0.01, *** *p* < 0.001.

**Figure 3 cancers-13-00756-f003:**
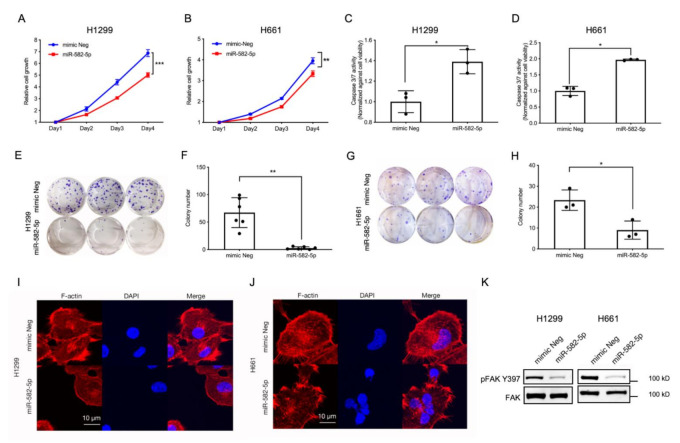
Ectopic expression of miR-582-5p inhibits the tumorigenic potential and actin cytoskeleton of NSCLC cells. (**A**,**B**) Cell proliferation was quantified following miR-582-5p overexpression in H1299 (**A**) and H661 (**B**) cell lines. (**C**,**D**) Relative cell apoptosis was measured in H1299 (**C**) and H661 (**D**) cell lines using a Caspase- 3/7 assay upon overexpression of miR-582-5p. The relative Caspase-3/7 activity was measured through the normalization of raw readouts against cell viability readouts that were estimated by CellTiter Glo. (**E**–**H**) Clonogenicity assays (**E**,**G**) were performed and stained colonies were quantified (**F**,**H**) using Image J software following the overexpression of miR-582-5p in H1299 (**E**,**F**) and H661 (**G**,**H**) cell lines. (**I**,**J**) Representative images of immunofluorescence staining for F-actin and DAPI following the overexpression of miR-582-5p in H1299 (**I**) and H661 (**J**) cells. Scale bar = 10 µm. (**K**) Western blot analysis of phosphorylated FAK (S397) and total FAK levels upon miR-582-5p overexpression in H1299 and H661 cells. Results are presented as the mean of 3 or 6 biological replicates with error bars representing standard deviation. Statistical analysis was conducted using Student’s t-test with * *p* < 0.05, ** *p* < 0.01, *** *p* < 0.001.

**Figure 4 cancers-13-00756-f004:**
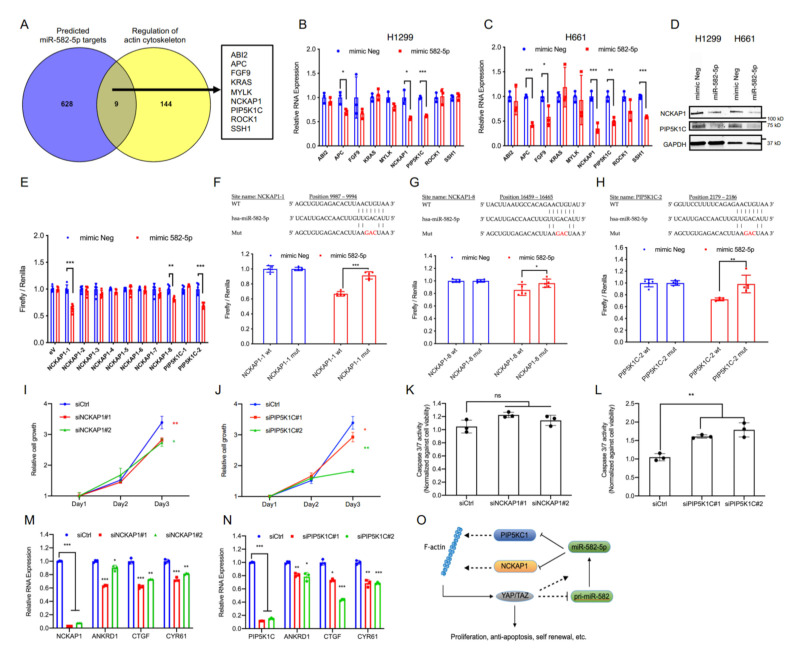
MiR-582-5p directly targets the 3’UTRs of NCKAP1 and PIP5K1C. (**A**) Venn diagram shows an integrated analysis of two gene lists, namely predicted miR-582-5p targets and regulators of actin cytoskeleton, which were derived from TargetScan and Wikipathways, respectively. Nine candidate genes were identified. (**B**,**C**) RT-qPCR analysis was performed to examine the changes in relative mRNA levels of the nine candidate genes post-transfection of miR-582-5p mimics in H1299 (**B**) and H661 (**C**) cells. The ΔCT of examined genes were normalized to that of 18SrRNA in order to determine the relative expression. (**D**) Western blot reflects the changes in protein levels of NCKAP1 and PIP5K1C in H1299 and H661 cells upon overexpression of miR-582-5p. (**E**) Dual luciferase assays were performed to evaluate the predicted binding sites of miR-582-5p to eight sites on NCKAP1 3’ UTR (NCKAP1-1—NCKAP1-8) and to two sites on PIP5K1C 3’UTR (PIP5K1C-1—PIP5K1C-2). Firefly luciferase readouts were normalized to the corresponding Renilla luciferase readouts to evaluate relative luciferase activity. (**F**–**H**) The wildtype (WT) and mutant (Mut) miR-582-5p binding sites on the 3’UTRs of NCKAP1 (**F**,**G**) and PIP5K1C (**H**) were constructed into a pmiR-Glo reporter plasmid followed by co-transfection of the aforementioned plasmids and microRNA mimics. Firefly luciferase readout was normalised to the corresponding Renilla luciferase readout. (**I**,**J**) Cell proliferation kinetics was quantified following the siRNA-mediated knockdown of NCKAP1 (**I**) and PIP5K1C (**J**). (**K**,**L**) Cell apoptosis was measured using Caspase-3/7 Glo upon the knockdowns of NCKAP1 (**K**) and PIP5K1C (**L**) in H1299 cells. The readouts of Caspase-3/7 Glo were normalized against that of CellTiter Glo. (**M**,**N**) RT-qPCR analysis was conducted upon siRNA-mediated knockdowns of NCKAP1 (**M**) and PIP5K1C (**N**) to examine the relative mRNA levels of NCKAP1, PIP4K1C, and the YAP/TAZ transcriptional targets, ANKRD1, CTGF, and CYR61 in H1299 cells. (**O**) A schematic working model illustrates the potential negative feedback loop between miR-582-5p and YAP/TAZ. YAP/TAZ represses pri-miR-582 gene expression but is eventually required for the maintenance of mature miR-582-5p expression. MiR-582-5p directly inhibits the expression of NCKAP1 and PIP5K1C and attenuates YAP/TAZ activity, potentially by compromising F-actin. Results are presented as the mean of three or five replicates as indicated with error bars representing standard deviation. Statistical analysis was conducted using Student’s t-test with * *p* < 0.05, ** *p* < 0.01, *** *p* < 0.001.

**Table 1 cancers-13-00756-t001:** List of primers’ sequences used in mRNA qRT-PCR.

Gene	Forward	Reverse
GAPDH	5′-TGGACTCCACGACGTACTCA-3′	5′-AATCCCATCACCATCTTCCA-3′
18SrRNA	5′-ACCCGTGGTCACCATGGTA-3′	5′-CGAACGTCTGCCCTATCAACTT-3′
YAP1	5′-AAGCTCAACTGAAGGCATGTCA-3′	5′-GGATCTCTAATGCAATGATAG-3′
TAZ (WWTR1)	5′-AACCCCAAGACTGAGGTGTG-3′	5′-TGACATCTCTCCGCTTCCTT-3′
ANKRD1	5′-TGCTGAATGCCTTCTCCCA-3′	5′-GCCTGCTGCCCTATCACA-3′
CTGF	5′-TGCAGTTCCTGACCCCTTAATG-3′	5′-AGCCAATTCCTGTAATGAACCAA-3′
KRAS	5′-GAGTACAGTGCAATGAGGGAC-3′	5′-CCTGAGCCTGTTTTGTGTCTAC-3′
PIP5K1C	5′-AGGCCATCGAATCGGATGAC-3′	5′-CCGAAAGACCGTGTTGCTCA-3′
NCKAP1	5′-TCCTAAATACTGACGCTACAGCA-3′	5′-GCCTCCTTGCATTCTCTTATGTC-3′
ITGB8	5′-ACCAGGAGAAGTGTCTATCCAG-3′	5′-CCAAGACGAAAGTCACGGGA-3′
SSH1	5′-ACCTTCTGCGTTGCGAAGAC-3′	5′-AGGTGGATTTTCGTGTCGCTC-3′
ABI2	5′-CAAAGCCTACACCACCCAATC-3′	5′-AGGTTGGCTGGAGCAATAATC-3′
ROCK1	5′-AAGTGAGGTTAGGGCGAAATG-3′	5′-AAGGTAGTTGATTGCCAACGAA-3′
MYLK	5′-CCCGAGGTTGTCTGGTTCAAA-3′	5′-GCAGGTGTACTTGGCATCGT-3′
FGF9	5′-GGCCTGGTCAGCATTCGAG-3′	5′-GTATCGCCTTCCAGTGTCCAC-3′
APC	5′-AAGCATGAAACCGGCTCACAT-3′	5′-CATTCGTGTAGTTGAACCCTGA-3′
pri-miR-582	5′-GTCATTCATGCACACATTGAAGAG-3′	5′-TCTACTAGAGAGAGATTTGCTAGTGGTGTT-3′

**Table 2 cancers-13-00756-t002:** Sequences of the primers and probe used to quantify 18SrRNA in miRNA RT-qPCR.

Gene	Sequences
18SrRNA	Forward primer: 5′-CGAACGTCTGCCCTATCAACTT-3′
	Reverse primer: 5’-ACCCGTGGTCACCATGGTA-3′
	Probe: 5′-FAM-TCGGAAGCTAAGCAGGGTCGGGC-Tamra-3′

**Table 3 cancers-13-00756-t003:** Details of primary antibodies used for Western blot. Cat # refers to Catalogue number.

Antibody	Antigen	Species	Dilution	Manufacturer	Cat #
Primary	YAP	Mouse	1:500	Santa Cruz	sc-101199
	TAZ	Rabbit	1:2000	Cell Signaling	4883S
	p-YAP (S127)	Rabbit	1:1000	Cell Signaling	4911S
	p-TAZ (S89)	Rabbit	1:1000	Cell Signaling	59971S
	CYR61	Rabbit	1:3000	Cell Signaling	14479S
	GAPDH	Rabbit	1:2000	Santa Cruz	sc-25778
	FAK	Rabbit	1:3000	Cell Signaling	13009S
	p-FAK Y397	Rabbit	1:1000	Cell Signaling	3283S
	NOTCH1	Rabbit	1:1000	Cell Signaling	4380S
	NOTCH2	Rabbit	1:1000	Cell Signaling	4530S
	NOTCH3	Rabbit	1:1000	Cell Signaling	5276S
	Pan-AKT	Rabbit	1:1000	Abcam	AB8805
	pLATS1(S909)	Rabbit	1:1000	Cell Signaling	9157S
	LATS1	Rabbit	1:1000	Protein Tech	17049-1-AP

**Table 4 cancers-13-00756-t004:** List of oligonucleotide sequences used in cloning.

Gene	Forward	Reverse
NCKAP1-1	5′-AATTCGACAGGAGCTGTGAGACACTTAACTGTAATCTTACC-3′	5′-TCGAGGTAAGATTACAGTTAAGTGTCTCACAGCTCCTGTCG-3′
NCKAP1-2	5′-AATTCCTTGTTTATTTCTGAAAAAGAACTGTATTTAGC-3′	5′-TCGAGCTAAATACAGTTCTTTTTCAGAAATAAACAAGG-3′
NCKAP1-3	5′-AATTCGAAACATTTGCCAAACTAAATACTGTAACACTGC-3′	5′-TCGAGCAGTGTTACAGTATTTAGTTTGGCAAATGTTTCG-3′
NCKAP1-4	5′-AATTCCAATGGCCTGATCTCGGCTCACTGTAACCTCCC-3′	5′-TCGAGGGAGGTTACAGTGAGCCGAGATCAGGCCATTGG-3′
NCKAP1-5	5′-AATTCGGAAATCATATTTAAAATTTACTGTAATTTTAC-3′	5′-TCGAGTAAAATTACAGTAAATTTTAAATATGATTTCCG-3′
NCKAP1-6	5′-AATTCGACTTTACATGTTTGATCTTGACTGTAAAACTAC-3′	5′-TCGAGTAGTTTTACAGTCAAGATCAAACATGTAAAGTCG-3′
NCKAP1-7	5′-AATTCGGCTTTCCTACTCAATCAAAAACTGTAGCTTGC-3′	5′-TCGAGCAAGCTACAGTTTTTGATTGAGTAGGAAAGCCG-3′
NCKAP1-8	5′-AATTCGAATGTACTTAATGCCACAGAACTGTATGCTTC-3′	5′-TCGAGAAGCATACAGTTCTGTGGCATTAAGTACATTCG-3′
NCKAP1-1mut	5′-GACAGGAGCTGTGAGACACTTAAGACTAATCTTACC-3′	5′-TCGAGGTAAGATTAGTCTTAAGTGTCTCACAGCTCCTGTCG-3′
NCKAP1-8mut	5′-AATTCGAATGTACTTAATGCCACAGAAGACTATGCTTC-3′	5′-TCGAGAAGCATAGTCTTCTGTGGCATTAAGTACATTCG-3′
PIP5K1C-1	5′-AATTCGCAGTGTCCAAATTCCTGTACTGTAAAGACTC-3′	5′-TCGAGAGTCTTTACAGTACAGGAATTTGGACACTGG-3′
PIP5K1C-2	5′-AATTCTTTGGTTCCTTTTCAGAGAACTGTAAACCG C-3′	5′-TCGAGCGGTTTACAGTTCTCTGAAAAGGAACCAAAG-3′
PIP5K1C-2mut	5′-AATTCTTTGGTTCCTTTTCAGAGAAGACTAAACCG C-3′	5′-TCGAGCGGTTTAGTCTTCTCTGAAAAGGAACCAAAG-3′
TAZ Subcloning	5′-CGTGGATCCGCCACCATGGACTACAAGACGATGACGACAAGAATCCGGCCTCGGCGCCCCCTC-3′	5′-CGTCTCGAGACGCGTTTACAGCCAGGTTAGAAAGG-3′
TAZ mutant	5′-CCGCTCGCACCGGTCGCCCGCG-3′	5′-ACATGCTGGGCACCCCCA-3′

## Data Availability

Publicly available datasets were analyzed in this study. These data can be found here: GSE161631.

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
