# Peer review of "miR-582-5p Is a Tumor Suppressor microRNA Targeting the Hippo-YAP/TAZ Signaling Pathway in Non-Small Cell Lung Cancer"

_cancers, 2021, doi:10.3390/cancers13040756_

Round 1

Reviewer 1 Report

 In this study, the authors identified miR-582-5p as a novel regulator of YAP/TAZ. Overall, the manuscript was well written, and the data quality is good. The storyline is also interesting, although it was damaged by the lack of sufficient evidence to support the conclusion. Herein, several major and minor concerns are listed below.

Major:

  1. The underlying mechanism by which YAP/TAZ maintains mature miRNA was not clearly addressed.
  2. Lack of rescue assay to support whether NCKAP1 and PIP5K1C are true targets of miR-582-5p in mediating its regulation on YAP and TAZ (i.e. phosphorylation and subcellular location). It would be more comprehensive to address it but not extend it in the future study as discussed.
  3. The function of miR-582-5p in tumorigenesis was not addressed by using animal models.

Other specific comments:

  1. How about the phosphorylation site of YAP and TAZ at S397 and S311?
  2. 1D, the authors described that miR-582-5p is positively correlated with the expression of p-YAP. Actually, it was the phosphorylation rate of YAP. A similar concern is applied to TAZ in Fig. 1F.
  3. Lack of direct evidence to indicate the nuclear YAP and TAZ is negatively correlated with miR-582-5p. The expression of nuclear YAP and TAZ should be determined.
  4. Supplementary Fig. 1 was not properly cited in the manuscript.
  5. In Fig. 2E, the cytoplasm location/nuclear location of YAP and TAZ should be quantitatively analyzed.
  6. Why the expression of CTGF was not changed in H661 (Fig. 2G) given it’s a classic target of YAP?

Reviewer 2 Report

The manuscript by Zhu et al, describes a role for miR-582-5p as a tumour suppressor in NSCLC by increasing levels of cytosolic/phosphorylated YAP/TAZ and decreasing mRNA levels of genes involved in focal adhesion/actin polymerisation. While the article presents a novel role for miR-582-5p in regulating Hippo-YAP/TAZ signalling in NSCLC I believe the data is somewhat preliminary and solid experimental evidence linking the two have not been presented.

Major concerns:

  • It is odd that the authors interrogated a gastric cancer RNAseq dataset, give the vast difference at the molecular level between gastric cancer and NSCLC. What was the rationale for this? Why was RNAseq not performed on NSCLC cell lines?
  • There seems to be a non-significant increase in pTAZ protein levels in miR-582-5p O/E H661 cell line (Fig 2D), given that the immunofluorescence/qPCR (Fig 2F/2H) also looks less than convincing, it does not fully support their conclusions. Furthermore, the authors should quantify the differences in sub-cellular localisation in the immunofluorescence images. While not entirely necessary, it would have been nice to demonstrate whether miR-582-5p increased association of pYAP/TAZ with 14-3-3 complexes.
  • The phenotypic data in Fig 3 looks convincing, although there is no direct association with Hipp-YAP/TAZ signalling. If these phenotypes are indeed due to decreased nuclear YAP/TAZ the authors should be able to rescue the phenotypes by overexpressing a S127A YAP and S89A TAZ mutant.
  • The link between the observable changes in actin dynamics to YAP/TAZ signalling and miR-582-5p again have not been shown. The authors should more clearly demonstrate this link.

Minor concerns:

  • The authors say the results demonstrated a negative correlation between miR-582-5p expression and nuclear YAP/TAZ, although the data they present demonstrates a positive correlation between cytosolic YAP/TAZ and miR-582-5p. While this may seem minor, the authors should not state results which have not been examined.
  • It would be nice to include the trendline in all scatterplots.
  • In 2.2 – last line, second paragraph – extend – change to extent.
  • In Table 3 (primary antibody concentrations) the concentration for pYAP (S89) is missing.

Round 2

Reviewer 1 Report

The authors have carefully addressed my questions, although the animal experimental evidence was not provided due to the limited turnaround time. 

Reviewer 2 Report

The reviewers have addressed all my previous concerns, and performed the essential rescue experiments required.